# Developing a Short-Form Buss–Warren Aggression Questionnaire Based on Machine Learning

**DOI:** 10.3390/bs13100799

**Published:** 2023-09-26

**Authors:** Xiuyu Jiang, Yitian Yang, Junyi Li

**Affiliations:** College of Psychology, Sichuan Normal University, Chengdu 610066, China; mrjiangxiuyu@outlook.com (X.J.); yyt200011@163.com (Y.Y.)

**Keywords:** aggression questionnaire, machine learning, short-form questionnaire, adolescents

## Abstract

For adolescents, high levels of aggression are often associated with suicide, physical injury, worsened academic performance, and crime. Therefore, there is a need for the early identification of and intervention for highly aggressive adolescents. The Buss–Warren Aggression Questionnaire (BWAQ) is one of the most widely used offensive measurement tools. It consists of 34 items, and the longer the scale, the more likely participants are to make an insufficient effort response (IER), which reduces the credibility of the results and increases the cost of implementation. This study aimed to develop a shorter BWAQ using machine learning (ML) techniques to reduce the frequency of IER and simultaneously decrease implementation costs. First, an initial version of the short-form questionnaire was created using stepwise regression and an ANOVA F-test. Then, a machine learning algorithm was used to create the optimal short-form questionnaire (BWAQ-ML). Finally, the reliability and validity of the optimal short-form questionnaire were tested using independent samples. The BWAQ-ML contains only four items, thirty items less than the BWAQ, and its AUC, accuracy, recall, precision, and F1 score are 0.85, 0.85, 0.89, 0.83, and 0.86, respectively. BWAQ-ML has a Cronbach’s alpha of 0.84, a correlation with RPQ of 0.514, and a correlation with PTM of −0.042, suggesting good measurement performance. The BWAQ-ML can effectively measure individual aggression, and its smaller number of items improves the measurement efficiency for large samples and reduces the frequency of IER occurrence. It can be used as a convenient tool for early adolescent aggression identification and intervention.

## 1. Introduction

### 1.1. Negative Effects of High Aggressiveness

Aggressive behavior is prevalent in children and adolescents. In a survey of approximately 15,000 middle school students in China, the detection rate of highly aggressive adolescents was 23.5% [1]; in a survey of approximately 80,000 middle school students in the United States, 18.8% of adolescents reported committing aggressive acts, and 20.1% of adolescents reported having been assaulted [2]. Additionally, highly aggressive individuals are more likely to maintain a high level of aggression. A short-term longitudinal study found that 87.4% of adolescents were still highly aggressive one year later compared to 52.3% of low-aggressive adolescents [3]. Aggressive behavior has many negative effects. High aggression can lead to physical injuries to others, with 700,000 adolescents (10–24 years old) in the United States alone being treated in hospitals for injuries from aggressive behavior in 2011 [4]. In addition, aggression is a potential risk factor for non-suicidal self-injury [5] and is positively associated with suicidal behavior [6,7]. Highly aggressive adolescents are more likely to commit crimes [8], and academic performance among them is worse [9,10,11]. Therefore, the efficient identification of highly aggressive adolescents and providing comprehensive interventions for aggressive adolescents are essential for reducing social risk and maintaining mental health.

### 1.2. Limitations of Buss–Warren Aggression Questionnaire

The Buss–Warren Aggression Questionnaire, which was developed based on the Buss–Perry Aggression Questionnaire (BPAQ) and contains a comparatively more concise and more transparent formulation of questions [12,13], is one of the most widely used instruments for measuring and rating aggression, and it has good psychometric properties [6,14]. However, the BWAQ consists of 34 items, and participants must exert a certain degree of effort to complete all of the items. In practice, participants do not answer every question carefully [15]. The incidence of insufficient effort response (IER) ranges up to 30% in all types of surveys [16]. As the length of the questionnaire increases, the proportion of participants seriously completing the questionnaire decreases, and the later the item, the worse the quality of the responses [17]. Therefore, the longer the questionnaire is, the more likely it is to result in more low-quality responses and lower questionnaire completion rates, making results less accurate. Failure to deal with this appropriately can adversely affect the veracity of study results [18]. An effective means of reducing IER is to reduce the length of the questionnaire; this is an ex ante control method that reduces the perceived difficulty of the task for participants, thereby reducing the frequency of IER [16]. Although shorter versions of the Aggression Questionnaire exist, such as the BPAQ-12 and BWAQ-15 [19,20], both shorter versions of the questionnaire have more than ten items and still require some effort for participants to complete. In addition, organizing and completing a large-scale measurement will inevitably consume many human and material resources for teachers, clinicians, and researchers, which may hamper research efforts. Therefore, it is necessary to use appropriate methods to streamline the questionnaire as much as possible while ensuring its performance. A streamlined questionnaire reduces IER and research costs, improves measurement efficiency, and enables the researcher to measure more variables simultaneously in one administration.

### 1.3. Developing Short Versions of Questionnaires Using Machine Learning

A common method researchers use in adapting and simplifying psychometric instruments is exploratory factor analysis, but this method requires items to be retained for each dimension of the questionnaire. Therefore, making psychometric instruments more streamlined via traditional methods is challenging. In recent years, some researchers have applied machine learning techniques to the study of simplifying psychometric tools. Machine learning belongs to the field of artificial intelligence, which can use the patterns learned from training data in the prediction of new data, and it is a reliable data analysis technique that can make the questionnaire simplification research free from the constraints of dimensions. An increasing number of studies have used machine learning techniques to simplify psychometric instruments, but they are mainly focused on clinical psychological assessment [21]. Wang reduced the Berg Balance Scale from 14 to 6 items based on machine learning techniques, with a 57% reduction in the number of items [22]. The R^2^ of the short version of the questionnaire was more significant than 0.96, and the LoAs were less than 95%, indicating good recognition performance. Lee used six machine learning algorithms to reduce the Insomnia Severity Index (which contains seven items) and Epworth Sleepiness Scale (which contains eight items) to a six-item short-form questionnaire, with a 60% reduction in the number of items, and accuracy reached 0.93 [23]. Orrù reduced the Structured Inventory of Malingered Symptomatology (which includes 75 items) to a 21-item short-form version, reporting a 72% reduction in the number of items and retaining 92% of the variance of the original scale [24]. Morrison reduced the Cognitive Distortions Questionnaire (which involves 15 items) into a 5-item ultrashort version, which was reduced by 67% from the original questionnaire, with an R^2^ of between 78.2 and 85.5% [25]. Lin reduced the Fugl-Meyer motor scale (which contains 50 items) into a short version of 10 items, a reduction of 80% in number, with Pearson’s correlation coefficient (r) ranging between 0.88 and 0.98 with the original measurement tool [26]. Machine learning techniques are a reliable method capable of simplifying psychometric instruments. However, past studies have primarily used R^2^ as performance indicators and have not explored the cutoff of short-form questionnaires. Machine learning classification algorithms can provide performance indicators such as AUC, accuracy, recall, precision, and F1 Score, which can help researchers to assess the optimal cutoff for short-form scales.

Therefore, this study proposes simplifying the BWAQ through the use of machine learning classification algorithms to develop a more streamlined short-form aggression questionnaire with an explicit cutoff to improve the efficiency of examining adolescent aggression and provide teachers, clinicians, and researchers with a more convenient tool.

## 2. Materials and Methods

### 2.1. Participants

Before the beginning of the survey, the investigator informed the participants that the purpose of the study was to understand the current situation of adolescents, that there are no right or wrong answers, that the results of the survey will be used only for scientific research, that the questionnaire was collected anonymously, and that the data were to be treated with strict confidentiality. The study was ethically approved by the Ethics Committee of Sichuan Normal University on 15 March 2023 (No. 2023LS029).

The participants were 796 middle school students. The sample was divided into two parts, namely simplification samples, which were used to simplify the BWAQ, and validation samples, which were used to validate the short-form BWAQ. There were 340 middle school students in the simplification samples, with a mean age of 14.83 ± 1.57 years. There were 200 female students, accounting for 58.8% of the simplification samples. The number of students categorized as highly aggressive by the BWAQ was 175 (or 51.5%). Seventh, eighth, tenth, and eleventh graders comprised 84, 75, 113, and 68 students, or 24.7%, 22.1%, 33.2%, and 20%, respectively. Validation samples comprised 456 middle school students with a mean age of 15.3 ± 1.3 years. There were 265 females, comprising 58.1% of the validation sample, and 236 students, i.e., 51.8% of the validation sample, whom the BWAQ-ML categorized as having highly aggressive tendencies. The percentages of grades 8 through 12 of the validation sample were 37.9%, 2.9%, 36.4%, 21.7%, and 1.1%, respectively.

### 2.2. Measurements

The BWAQ has 34 items belonging to the following five dimensions: physical aggression, verbal aggression, hostility, anger, and indirect aggression. The scale consists of 5 points, with “1” indicating “not at all like me” and “5” indicating “completely like me”, with higher scores indicating higher levels of aggression. Maxwell revised the Chinese version of the BWAQ [27], based on which Zhang developed the standard BWAQ for students aged 12 to 18 years old, with total scores of 89 and above potentially indicating highly aggressive behavior [28,29].

The validity scale adopted for this study used the Reactive–Proactive Aggression Questionnaire (RPQ) developed by Raine [30]. The RPQ is divided into two dimensions: reactive aggression (which includes 11 items) and proactive aggression (which includes 12 items). It is scored on a 3-point scale: 0 for never, 1 for sometimes, and 2 for often. Higher total RPQ scores indicate higher levels of aggression. The RPQ has good psychometric properties in mainland China [31,32,33].

Carlo and Randall developed the Prosocial Tendencies Measure (PTM) to measure individuals’ prosocial tendencies [34], and we used the PTM to test the discriminant validity of the short-form scale. The PTM consists of 23 items and is scored on a 5-point, with “1” meaning “not at all like me” and “5” meaning “completely like me”. Higher total PTM scores indicate higher prosocial tendencies. The PTM has good psychometric properties in mainland China [35].

### 2.3. Statistical Analysis

We analyzed data using Python 3.9 in PyCharm 2023.1.2 (Community Edition). The whole process was divided into two phases: simplification and validation. In the simplification phase, the simplification sample is randomly divided into a training set (80% simplification sample) that has been used for feature selection and model training and a test set (20% simplification sample) to evaluate the model’s performance. The first step is feature selection. The initial version of the short-form BWAQ was created by analyzing the training set using stepwise regression and an ANOVA F-test to extract the items that contribute most to the questionnaire results (i.e., the most important features) [36,37]. The second step is machine learning modeling. The initial version of the short-form BWAQ was modeled by classification algorithms, including a Logistic Regression (LR) algorithm, a Support Vector Machine (SVM), a Random Forest (RF) algorithm, and a Naïve Bayes (NB) algorithm [38,39]. Then, the short-form BWAQ was evaluated according to the model’s AUC, accuracy, recall, and precision performance to construct the optimal short-form questionnaire. The third step is to determine the cutoff. We calculated AUC, accuracy, recall, precision, and F1 scores based on the confusion matrix under different cutoff conditions to specify the cutoff of BWAQ-ML. The cutoff is the minimum score at which a participant is categorized as highly aggressive. During the validation phase, we verified the reliability and validity of BWAQ-ML using validation samples, and we tested the reliability of the BWAQ-ML as a streamlined version of the BWAQ using RPQ and PTM.

The specifics of the evaluation metrics for the machine learning models are described below:(1)The horizontal axis of the ROC curve is the False Positive Rate, and the vertical axis is the True Positive Rate. AUC is the area under the ROC curve; its value ranges from 0 to 1. The closer the AUC is to 1, the more correctly the model can distinguish between positive and negative cases. The calculation method of AUC considers both the classification ability of the classifier for positive and negative cases. It is still able to make a reasonable evaluation of the classifier in cases of sample imbalance. Therefore, AUC can be regarded as the primary index for evaluating the classification ability of a model [38].(2)Accuracy indicates the proportion of the whole dataset that a model correctly classifies. Accuracy has the advantage of being easy to understand and facilitates communication for non-technical people, but accuracy may not be effective enough in unbalanced datasets. For example, a model will be highly accurate in a dataset with far more negative than positive cases, even if it classifies all the data as negative. Therefore, other metrics are often calculated when evaluating the performance of a machine learning model [38].(3)Recall, also known as True Positive Rate, indicates the percentage of positive case samples that the model correctly predicts. A higher recall indicates that the model can better identify positive case samples but may also result in more false positive cases [38].(4)Precision measures the proportion of samples predicted by the model to be positive cases that are true positive cases. Precision concerns how many of the model’s predictions of positive examples are correct. Thus, recall and precision are complementary [38].(5)The F1 score is the weighted average of recall and precision. In cases where both recall and precision need to be taken into account, the closer the F1 score is to 1, the better balance the model achieves between recall and precision, and the model’s comprehensive performance is better [38].

## 3. Results

### 3.1. Simplifying BWAQ

We determined the initial version of the short-form questionnaire using stepwise regression and an ANOVA F-test. At a significance level of 0.01, stepwise regression identified the eight most important features, which (in descending order of importance) were items 29, 14, 21, 18, 7, 30, 1, and 23 of the BWAQ. The ANOVA F-test ranked the importance of the features differently than the stepwise regression (Table 1). In the ANOVA F-test, the larger the F-value of a feature is, the higher its correlation with the outcome, and the *p*-value is the significance level of this correlation. Overall, 32 of all of the 34 items have a significance level below 0.001, and the top 25% are the most important features according to our collected F-values, and they are (in descending order) items 29, 7, 21, 9, 12, 5, 33, and 14 of the BWAQ. Stepwise regression and the ANOVA F-test each identified eight short-form questionnaires of different lengths (one to eight items), resulting in sixteen initial short-form questionnaires. These initial questionnaires were named after the algorithms and the number of items they contained. For example, AF-4 is the short-form questionnaire consisting of the first four items of the eight items identified using the ANOVA F-test. In comparison, SR-8 is the short-form questionnaire consisting of all eight items of the eight items identified using stepwise regression.

The modeling results of the 16 initial short-form questionnaires generated via a Logistic Regression, Support Vector Machine, Random Forest algorithm, and Naïve Bayes algorithm showed that the AUC of all models ranged from 0.72 to 0.98 (Table 2), indicating that all the initial short-form questionnaires are effective in recognizing individual aggression. In general, the trained machine learning models perform better on the training set than the test set, and the smaller the difference in AUC between the models on the training set and the test set, the more stable their performance is and the closer their prediction results are to the actual results. We further compared the differences in the AUC of different algorithms on training and test data (Figure 1) and found that the AUC of the Random Forest algorithm is the most stable among all algorithms, making it more suitable for subsequent analyses.

As can be seen in Table 3, among the sixteen initial versions of the short-form questionnaire models in Random Forest, there are three short-form questionnaires with AUC, accuracy, recall, and precision values above 0.8, namely SR-5, AF-4, and AF-8 (Table 3). AF-4 has 50% fewer items than AF-8; AUC, accuracy, and precision have decreased by 4%, 1%, and 4%, respectively, and recall has increased by 3%. AF-4 has 20% fewer items than SR-5, 1% less AUC, and the same accuracy, recall, and precision. In contrast, AF-4 reduced the number of items by 20–50%, but the performance indicator changed by only 1–5%, and there was a high correlation with BWAQ (Pearson’s correlation = 0.85; *p* < 0.001).

AF-4 includes four items, and its cutoff ranges from 4 to 20. We calculated the evaluation metrics corresponding to each cutoff according to the confuse matrix (Table 4). As the cutoff increases, AUC and accuracy first increase and then decrease; recall decreases from 1 to 0.04, and precision increases from 0.52 to 1 (Figure 2). When the cutoff of the AF-4 is changed, the relationship between recall and precision becomes a trade-off relationship. Therefore, the performance of the questionnaire under a particular cutoff condition cannot be judged by either recall or precision alone; both recall and precision are essential and need to be considered. Therefore, to better estimate the cutoff, it is necessary to calculate the F1 score, which is the harmonic mean of recall and precision, to estimate the optimal cutoff for the AF-4 (Table 4). Comparing the AF-4 under 17 cutoff conditions, AF-4 has the best AUC, accuracy, and F1 score when the cutoff is 11. Therefore, the optimal short-form version of BWAQ based on the machine learning algorithm is AF-4 with a cutoff of 11, which we named BWAQ-ML.

### 3.2. Validating BWAQ

Cronbach’s alpha for the BWAQ-ML, RPQ, and PTM were 0.84, 0.83, and 0.93, respectively, and Pearson’s correlation for the BWAQ-ML with the RPQ and PTM was 0.514 (*p* < 0.001) and −0.042 (*p* = 0.375), respectively, indicating that the BWAQ-ML can effectively measure individual aggression. We labeled samples scoring in the top 50% on the RPQ as high aggression and those scoring in the bottom 50% as low aggression. Comparing the classification results of the BWAQ-ML and the RPQ, the degree of overlap was 71%, indicating that the content measured by the BWAQ-ML was consistent with that of the RPQ. There was a significant positive correlation between the items of the BWAQ-ML, with correlation coefficients ranging from 0.469 to 0.621; there was a significant positive correlation between the items and the total score, with correlation coefficients ranging from 0.618 to 0.730 (Table 5), which indicates good differentiation between the items.

## 4. Discussion

The number of items in the BWAQ-ML is only 12% of the BWAQ but retains 85% of the AUC and accuracy, 89% of the recall, and 83% of the precision of the full version of the questionnaire. In other words, the BWAQ-ML, despite having an 88% reduction in the number of items, can identify 89% of the cases classified by the BWAQ as high aggression. The internal consistency coefficient of the short version of the questionnaire was 0.84, which was significantly positively correlated with the RPQ, and its categorization results reached a 71% overlap with the RPQ, with no significant correlation with the PTM, which indicated that the BWAQ-ML had good reliability and validity. Overall, the BWAQ-ML, which contains only four items, has sufficient validity as a condensed version of the BWAQ.

Previous studies have used regression algorithms to develop a shortened version of the scale without a cutoff (or call it a threshold) [22,24,25,26], making it difficult for teachers and clinicians to categorize participants as positive or negative cases based on the shortened version of the scale alone. In contrast, the BWAQ-ML is a shortened version of the BWAQ that has been developed based on machine learning classification algorithms with an explicit cutoff, and the results that can be derived from its use are easier to understand and interpret. In practice measurements, teachers and clinicians can quickly identify positive examples from the survey sample (i.e., individuals with a BWAQ-ML score of no less than 11). Therefore, for teachers and clinicians, the BWAQ-ML dramatically improves the test’s efficiency and makes screening for highly aggressive individuals easy to administer. The BWAQ-ML helps identify highly aggressive youths early and focuses more resources on those who need further attention through interventions that can reduce the risk of academic underachievement, violent victimization, delinquency, suicide, and self-injury.

As the length of the questionnaire increases, participants experience IER due to carelessness or a lack of effort [15], and the more complex the items, the more likely they are to experience IER [17], which can adversely affect the credibility of the study results [18]. The percentage of participants experiencing IER while completing the questionnaire can be up to 30% [16]. Therefore, the questionnaire should not be too long. The BWAQ-ML contains only four items, and the time required for participants to complete all the items is no more than 1 min, which significantly reduces the perceived difficulty of the participants in completing the questionnaire, improves the completion rate and credibility of the questionnaire responses, and reduces the likelihood of the emergence of IER. In addition, by using the short-form questionnaire, the researcher could also measure more variables at once during the administration process. If necessary, the researcher can obtain the results quickly via oral calculation on the spot, which is easy and minimizes the risk of errors. For researchers, the BWAQ-ML (4-item) has two advantages over the BWAQ (34-item questionnaire): Firstly, respondents will be less likely to experience IER while completing the BWAQ-ML. Secondly, the researcher can administer more scales while ensuring accuracy, which could help to improve the efficiency of studies. In addition, the data output from the BWAQ-ML can be either categorical or continuous, which also meets different needs in the data analysis process.

The BWAQ contains five aggressive behavioral manifestations: physical aggression, verbal aggression, hostility, anger, and indirect aggression [13]. The first two items of the BWAQ-ML belong to the anger dimension, and the last two belong to the hostility dimension. The positive correlation between hostility and suicide is more significant than the other dimensions [6,40]. Therefore, the short-form questionnaire may be more conducive to detecting highly aggressive individuals with suicidal tendencies. In addition, since these four items do not directly ask participants about the frequency of aggression, they reduce the social desirability of participants, which would facilitate the investigator’s early identification of high-risk individuals for comprehensive intervention.

This study has the following limitations. First, the BWAQ-ML is simplified based on the BWAQ, so the performance of the short-form questionnaire relies on the original questionnaire. Second, the mean score of the BWAQ’s norm for adolescents aged 12 to 18 ranged from 61 to 79, but the mean score for the simplification samples in this study was 90. There are two possible reasons for the higher mean score: (1) the urban norms were developed 12 years ago (2011); (2) this study was at the end of the secondary school year when the sample was administered, and exam pressure may have influenced subjects’ scores on the BWAQ. Third, although Random Forest, which has a more stable AUC, is used as a reliable algorithm in the simplification phase, it is still possible that the machine learning algorithm may be overfitted due to the small sample size, affecting the results. Fourth, the BWAQ-ML has only four items, which is unsuitable for embedding recognition scales in the questionnaire for the recognition of IER. In summary, it is recommended that future studies validate the BWAQ-ML using a larger sample.

## 5. Conclusions

The BWAQ-ML contains four items, each with a score range of 1 to 5, and respondents are categorized as highly aggressive when they score 11 and above. The number of items in the BWAQ-ML is 12% of the BWAQ, and its predictive performance reaches 83-89% (AUC = 0.85, accuracy = 0.85, recall = 0.89, precision = 0.83) of the BWAQ. Validation analyses using independent samples indicate that the BWAQ-ML has good reliability (Cronbach’s alpha = 0.84) and validity (correlation with RPQ is 0.514, *p* < 0.001; correlation with PTM is −0.042, *p* = 0.375). In other words, the BWAQ-ML, as a streamlined version of the BWAQ, substantially reduces the number of items and has better measurement performance.

The applications of the BWAQ-ML are promising. Firstly, aggressive adolescents are at risk of academic failure, physical injury, delinquency, suicide, and self-harm, and the BWAQ-ML makes the measurement of aggression easy to administer, which can help to identify highly aggressive adolescents at an early stage, increasing the potential for intervention. Secondly, reducing the length of the questionnaire is an effective means of reducing IER [16], and the BWAQ-ML, which contains only four items, reduces the frequency of IER among participants. In addition, its streamlined nature allows researchers to measure more variables while ensuring accuracy, which could help to improve the efficiency of studies. Third, the BWAQ-ML has clear thresholds that measure aggression as simple and easy to understand and interpret, reducing the workload of clinical workers and teachers, saving costs, and making measuring aggression more efficient.

The BWAQ-ML is a streamlined version of the BWAQ that can be used to measure adolescent aggression effectively. The items of BWAQ-ML are as follows:(1)At times I feel like a bomb ready to explode.(2)At times I get very angry for no good reason.(3)I sometimes feel that people are laughing at me behind my back.(4)I wonder why sometimes I feel so bitter about things.

## Figures and Tables

**Figure 1 behavsci-13-00799-f001:**
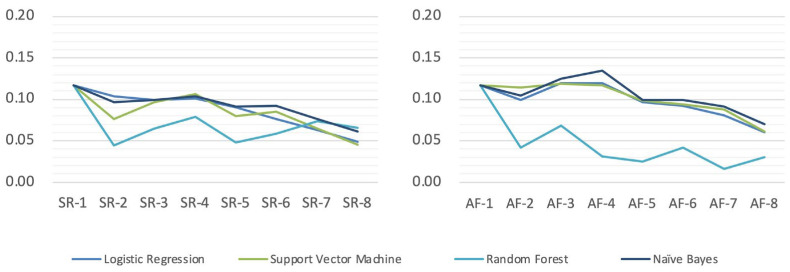
AUC differences between machine learning algorithms on training and test sets.

**Figure 2 behavsci-13-00799-f002:**
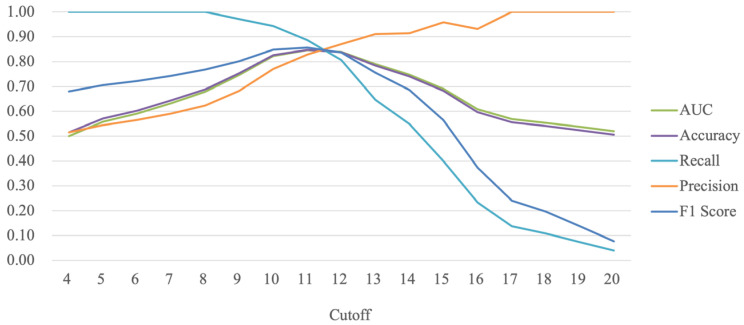
Trend of the evaluation metrics with cutoff.

**Table 1 behavsci-13-00799-t001:** Ranking of feature importance using ANOVA F-tests.

Item Number of BWAQ	F-Value	*p*-Value
29	144.42	0.000
7	139.02	0.000
21	121.43	0.000
9	113.42	0.000
12	111.52	0.000
5	107.33	0.000
33	107.04	0.000
14	101.99	0.000
32	95.03	0.000
31	89.48	0.000
22	85.48	0.000
23	78.79	0.000
11	76.98	0.000
17	74.72	0.000
13	67.59	0.000
30	65.80	0.000
16	65.04	0.000
20	61.14	0.000
8	57.78	0.000
1	57.73	0.000
4	55.79	0.000
10	55.43	0.000
34	53.03	0.000
6	53.00	0.000
15	47.13	0.000
2	46.40	0.000
3	45.77	0.000
18	43.27	0.000
25	43.24	0.000
27	40.16	0.000
24	36.40	0.000
28	19.15	0.000
26	6.53	0.011
19	4.53	0.034

**Table 2 behavsci-13-00799-t002:** AUC of machine learning algorithms on initial version of short-form questionnaires.

Questionnaire	Logistics Regression		Support Vector Machine		Random Forest		Naïve Bayes	
	Train	Test	Train	Test	Train	Test	Train	Test
SR-1	0.83	0.72	0.83	0.72	0.83	0.72	0.83	0.72
SR-2	0.90	0.79	0.90	0.82	0.87	0.83	0.89	0.79
SR-3	0.93	0.83	0.92	0.83	0.87	0.81	0.92	0.83
SR-4	0.94	0.84	0.94	0.84	0.90	0.82	0.93	0.83
SR-5	0.95	0.86	0.95	0.87	0.92	0.87	0.94	0.85
SR-6	0.97	0.89	0.97	0.88	0.94	0.88	0.96	0.87
SR-7	0.98	0.91	0.98	0.91	0.95	0.87	0.97	0.89
SR-8	0.98	0.93	0.98	0.93	0.95	0.89	0.97	0.91
AF-1	0.83	0.72	0.83	0.72	0.83	0.72	0.83	0.72
AF-2	0.88	0.78	0.89	0.78	0.86	0.82	0.89	0.78
AF-3	0.93	0.81	0.92	0.80	0.85	0.79	0.93	0.80
AF-4	0.94	0.82	0.94	0.83	0.89	0.86	0.95	0.81
AF-5	0.95	0.85	0.95	0.85	0.90	0.88	0.95	0.85
AF-6	0.95	0.85	0.95	0.85	0.90	0.86	0.95	0.85
AF-7	0.95	0.87	0.96	0.87	0.93	0.91	0.95	0.86
AF-8	0.96	0.90	0.96	0.90	0.93	0.90	0.96	0.89

**Table 3 behavsci-13-00799-t003:** Performance of the initial version of short-form questionnaires based on Random Forest.

Questionnaire	AUC	Accuracy	Recall	Precision
SR-1	0.72	0.77	0.85	0.77
SR-2	0.83	0.77	0.85	0.77
SR-3	0.81	0.72	0.80	0.74
SR-4	0.82	0.75	0.78	0.80
SR-5	0.87	0.81	0.83	0.85
SR-6	0.88	0.79	0.78	0.86
SR-7	0.87	0.81	0.78	0.89
SR-8	0.89	0.79	0.75	0.88
AF-1	0.72	0.77	0.85	0.77
AF-2	0.82	0.75	0.75	0.81
AF-3	0.79	0.75	0.78	0.80
AF-4	0.86	0.81	0.83	0.85
AF-5	0.88	0.77	0.75	0.83
AF-6	0.86	0.77	0.75	0.83
AF-7	0.91	0.81	0.78	0.89
AF-8	0.90	0.82	0.80	0.89

**Table 4 behavsci-13-00799-t004:** Evaluation metrics for different cutoffs of AF-4.

Cutoff	AUC	Accuracy	Recall	Precision	F1 Score
4	0.50	0.52	1.00	0.52	0.68
5	0.56	0.57	1.00	0.55	0.71
6	0.59	0.60	1.00	0.57	0.72
7	0.63	0.64	1.00	0.59	0.74
8	0.68	0.69	1.00	0.62	0.77
9	0.75	0.75	0.97	0.68	0.80
10	0.82	0.83	0.94	0.77	0.85
11	0.85	0.85	0.89	0.83	0.86
12	0.84	0.84	0.81	0.87	0.84
13	0.79	0.79	0.65	0.91	0.76
14	0.75	0.74	0.55	0.91	0.69
15	0.69	0.68	0.40	0.96	0.56
16	0.61	0.60	0.23	0.93	0.37
17	0.57	0.56	0.14	1.00	0.24
18	0.55	0.54	0.11	1.00	0.20
19	0.54	0.52	0.07	1.00	0.14
20	0.52	0.51	0.04	1.00	0.08

**Table 5 behavsci-13-00799-t005:** Correlation matrix for each item and total of BWAQ-ML.

	BWAQ-ML_1	BWAQ-ML_2	BWAQ-ML_3	BWAQ-ML_4	BWAQ-ML_Total
BWAQ-ML_1	1				
BWAQ-ML_2	0.579 **	1			
BWAQ-ML_3	0.469 **	0.547 **	1		
BWAQ-ML_4	0.609 **	0.621 **	0.609 **	1	
BWAQ-ML_Total	0.618 **	0.702 **	0.677 **	0.730 **	1

Note: ** Correlation is significant at the 0.01 level (two-tailed).

## Data Availability

The data presented in this study are openly available in OSF Registries at https://osf.io/znr46/ (accessed on 29 July 2023).

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
