# Peer review of "Developing a Short-Form Buss–Warren Aggression Questionnaire Based on Machine Learning"

_behavsci, 2023, doi:10.3390/bs13100799_

Round 1
Reviewer 1 Report
1. This is a meaningful study as this article contributes to furthering the analysis of developing a short-form buss-warren aggression questionnaire for early identification and intervention for highly aggressive adolescents.
2. The title and abstract are appropriate for the content of the text.
3. The article is well constructed.
4. The analysis was well performed.
5. The conclusion basically responds to the research purpose. The conclusion section should be more than just a few lines of sentences of the questionnaire. The conclusion could be strengthened
6. The reviewer suggests including a brief description in the conclusion section of how the findings of this study can be applied in machine learning for early identification and intervention for highly aggressive adolescents.
None
Author Response
Thank you for your review of our paper. We have addressed each of your points below.
- This is a meaningful study as this article contributes to furthering the analysis of developing a short-form buss-warren aggression questionnaire for early identification and intervention for highly aggressive adolescents.
- The title and abstract are appropriate for the content of the text.
- The article is well constructed.
- The analysis was well performed.
- The conclusion basically responds to the research purpose. The conclusion section should be more than just a few lines of sentences of the questionnaire. The conclusion could be strengthened.
Response: We have strengthened the conclusions, which can be found in Section 5 ('Conclusion', page 10, lines 330-349).
- The reviewer suggests including a brief description in the conclusion section of how the findings of this study can be applied in machine learning for early identification and intervention for highly aggressive adolescents.
Response: We have added to the conclusions section a note that the scales developed by the machine learning classification algorithm are easy for teachers and clinicians to work with and a contextualized account of how to identify highly aggressive adolescents and intervene early with them. This part has been revised in Section 5 ('Conclusion', page 10, lines 330-349).

Reviewer 2 Report
Nice article, clearly presented and highly valuabe cotribution.
I would potentially describe better the differfencers/definitions of AUC, cutt-off, Accuracy, Recall, Precision, as they are key for developing the shortened versions.
Also, the background/references for specific threshold values would need to be better described.
Author Response
Thank you for your review of our paper. We have addressed each of your points below. In doing so, we have labeled your original comments for a point-by-point response.
- I would potentially describe better the differfencers/definitions of AUC, cutt-off, Accuracy, Recall, Precision, as they are key for developing the shortened versions.
Response: The definition of cutoff has been presented in Section 2.3 ('Statistical Analysis', page 4, lines 159-160). In addition, we have added explanations for AUC, accuracy, recall, precision, and F1 Score presented in Section 2.3 ('Statistical Analysis', page 4, lines 164-191).
- Also, the background/references for specific threshold values would need to be better described.
Response: In determining the cutoff for the short-form BWAQ, we considered all possible values of cutoff simultaneously (4 items, 5-point scoring, and a range of possible cutoff values from 4 to 20). We calculated AUC, accuracy, recall, precision, and F1 Score for different cutoffs. In contrast, the short-time scale performed best when the cutoff was 11. Therefore, we identified 11 as the threshold for the short-time scale (i.e., respondents were classified as highly aggressive when they scored greater than or equal to 11 on the short-time scale). The exact process of selecting the threshold has been showed in Section 3.1 ('Simplifying BWAQ', page 7, lines 236-248).

Reviewer 3 Report
This is a very well-written paper. It clearly explains the topic. My only concern is the extremely short Conclusion part, which makes no sense to me. It should be rewritten to support the article and provide a vivid picture of your conclusions.
In order to elaborate further please address the following questions:
What are the conclusions of the survey? Do the results from the survey answer the original research question? How do the authors support their conclusions? Please provide specific examples from the findings. Are the conclusions logical and solid or do they need further examination? Comparison with other studies. What other questions might arise for future research?
Author Response
Thank you for your review of our paper. We have addressed each of your points below. In doing so, we have labeled your original comments for a point-by-point response.
- What are the conclusions of the survey? Do the results from the survey answer the original research question? How do the authors support their conclusions? Please provide specific examples from the findings.
Response: We have rewritten the conclusions, addressing the support of the conclusions by specific findings, and responding to each of the initial research questions. These have been revised in Section 5 ('Conclusions', page 10, lines 330-349).
- Are the conclusions logical and solid or do they need further examination? Comparison with other studies.
Response: Compared to short-form scales developed by regression algorithms, our short-form scales developed based on categorization algorithms have an explicit cutoff, which significantly facilitates the administration of the scales by clinicians, teachers, and researchers. This part has been revised in Section 4 (Discussion, page 9, lines 276-288).
- What other questions might arise for future research?
Response: Research related to the development of short-form scales using machine learning algorithms has just begun in recent years, so there is less relevant empirical evidence, and the applicability of the method in the field of psychometrics needs to be demonstrated by psychometric experts in future studies. This study describes the research limitations in the discussion and points out the problems and suggestions that may be encountered in future research. The part has been presented in Section 4 ('Discussion', page 9, lines 315-328).

Reviewer 4 Report
Dear Sir/Mam
Please find bellow the requested review regarding the manuscript. The article contains a lot of useful information on the issue. The topic is very interesting and but use of sources is not appropriate. Although it has some useful information there are less references and the statements are not established. I suggest the authors to write more information with references.
The article contains a lot of useful information on the issue. It is quite clear what is already known about this topic and the research question is clearly outlined. The abstract is too brief and introduction section involves too many information. The research question is not justified clearly, given what is already known about the topic. The results are not discussed from multiple angles and conclusions answer the aims of the study partially. The conclusions are partially supported by references or results and the limitations of the study fatal and it is questionable if there are opportunities to inform future research. Positive: There are some strengths of the article that could have an impact in the field, such as the topic and its impact on the existed literature. The manuscript is approved publication only after major changes.
Minor changes
Author Response
Thank you for your review of our paper. We have addressed each of your points below. In doing so, we have labeled your original comments for a point-by-point response.
- Although it has some useful information there are less references and the statements are not established. I suggest the authors to write more information with references.
Response: We have added references to give strong support to our statements.
(1) We supplemented the information on Feature Selection Algorithms by adding references to ANOVA F-test and Stepwise Regression, as detailed on Section 2.3 ('Statistical Analysis', page 4, lines 150-152).
(2) We supplemented the information on Machine Learning Classification Algorithms by adding references to Random Forest, Support vector machine, Naïve Bayes, and Logistic Regression, as detailed on Section 2.3 ('Statistical Analysis', page 4, lines 153-157).
(3) We supplemented the information on model evaluation metrics by adding references to AUC, Accuracy, Recall, Precision, and F1 score, as presented on Section 2.3 ('Statistical Analysis', page 4, lines 164-191).
- The article contains a lot of useful information on the issue. It is quite clear what is already known about this topic and the research question is clearly outlined. The abstract is too brief and introduction section involves too many information.
Response: We have added content to the abstract detailed on page 1 (lines 9-10; lines 17-24). We have also added subheadings to the introduction section to make the presentation of information more logical and easier to read, to avoid information being read in a miscellaneous, detailed on pages 1 (line 28) and page 2 (line 45, line 70).
- The research question is not justified clearly, given what is already known about the topic. The results are not discussed from multiple angles and conclusions answer the aims of the study partially.
Response: We have reworked the discussion section to discuss the findings from multiple perspectives, responding to all the research aims in the context of the rationale for studying the problem and illustrating its usefulness for future research with examples. Please refer to Section 4 ('Discussion', page 9, lines 276-305) and Section 5 ('Conclusions', page 10, lines 330-349) for more details.
- The conclusions are partially supported by references or results and the limitations of the study fatal and it is questionable if there are opportunities to inform future research.
Responese:
(1) We reworked the Section 5 ('Conclusions', page 10, lines 330-349) to precisely describe the link between the results and conclusions. We added content and references to the Section 4 ('discussion', page 9, lines 276-288) to increase the support of the findings to the study's conclusions.
(2) Studies on developing short-form scales using machine learning algorithms have only begun in recent years. They are still at an early stage, with less empirical evidence. However, related studies affirm the contribution of machine learning techniques to simplifying scale research, this part is in Section 1.3 ('Machine learning algorithms can develop short versions of questionnaires', page 2, lines 70-100). Our study can not only provide methodological references for the field but also increase the empirical evidence in the field. Regarding the study's limitations, we have carefully considered your comments and responded as follows.
Limitation 1: Studies of simplified scales have been conducted based on the original scale, and we are no exception. Future studies may consider revisions based on the short-term scales to improve their measurement performance.
Limitation 2: The same questionnaire given to different groups of people at different times will have differences. Our research conclusions were drawn by measuring and analyzing the current group of people, and the validation analysis of independent samples showed that the BWAQ-ML has good psychometric properties, indicating that the conclusions are reliable. We believe that future research can build on the findings of this study by updating the normative data.
Limitation 3: Sample size may affect machine learning algorithms, but sample size can also affect other methods of simplifying scales. We used four machine learning algorithms to model the data, and the slight differences in predictions between different models and between different datasets of the same model and the validation analyses support the fact that the BWAQ-ML has good psychometric properties, which suggests that the findings are reliable. We believe that future studies could validate the BWAQ-ML using larger samples.
Limitation 4: Although the BWAQ-ML is not suitable for inserting IER identification questions (as this would be too easy to notice), given that it is so brief, the number of instances in which participants appear to be not answering carefully would be significantly reduced, and its practical implications are significant.
In summary, although our study has some limitations, the findings are supported by the results of the data analysis, and the conclusions are reliable. We suggest that future studies could further validate the BWAQ-ML using larger sample data; also, psychometric experts need to demonstrate the practicality of machine learning techniques in psychometrics in future studies.

Round 2
Reviewer 4 Report
Agreed